# Studies of Utilization of Technogenic Raw Materials in the Synthesis of Cement Clinker from It and Further Production of Portland Cement

Nurgali Zhanikulov [1], Bayan Sapargaliyeva [2], Aktolkyn Agabekova [3], Yana Alfereva [4], Aidin Baidibekova [5], Samal Syrlybekkyzy [6,*], Lazzat Nurshakhanova [7], Farida Nurbayeva [6], Gulzhan Sabyrbaeva [6], Yergazy Zhatkanbayev [8], Pavel Kozlov [9,*], Aizhan Izbassar [10] and Olga Kolesnikova [11,*]

1   Academician E.A. Buketov Karaganda University, Karaganda 100028, Kazakhstan; znn@yandex.kz
2   Abai Kazakh National Pedagogical University, Almaty 050010, Kazakhstan; bonya_sh@mail.ru
3   Department of Electrical Engineering, H. A. Yassavi International Kazakh-Turkish University, Turkestan 161200, Kazakhstan; aktolkyn_agabekova@mail.ru
4   M.V. Lomonosov Moscow State University, 119991 Moscow, Russia; yana.rs@bk.ru
5   Department of Higher Mathematics and Physics for Technical Specialties, M. Auezov South Kazakhstan University, Shymkent 160012, Kazakhstan; a.baidibekova@mail.ru
6   Department of Ecology and Geology, Sh. Yesenov Caspian University of Technology and Engineering, Aktau 130002, Kazakhstan; farida.nurbayeva@yu.edu.kz (F.N.); sabyrbaeva@yandex.kz (G.S.)
7   Department of Petrochemical Engineering, Sh. Yesenov Caspian University of Technology and Engineering, Aktau 130002, Kazakhstan; lazzat.nurshakhanova@yu.edu.kz
8   Department of Chemistry, M. Auezov South Kazakhstan University, Shymkent 160012, Kazakhstan; ergazy-1980@mail.ru
9   Polytechnic Institute, Far Eastern Federal University, 690922 Vladivostok, Russia
10   Department of Construction Engineering, Sh. Yesenov Caspian University of Technology and Engineering, Aktau 130002, Kazakhstan; aizhan.izbassar@yu.edu.kz
11   M. Auezov South Kazakhstan University, Shymkent 160012, Kazakhstan
*   Correspondence: samal.syrlybekkyzy@yu.edu.kz (S.S.); opiv.uvc@dfu.ru (P.K.); ogkolesnikova@yandex.kz (O.K.); Tel.: +77-05-2566-897 (O.K.)

**Abstract:** Four series of experiments were carried out to study the possibility of replacing clay and an iron-containing component with tefritobasalt and lead slag as part of the initial charge for Portland cement. The experiments were carried out at atmospheric pressure and a temperature of 1350 °C. It was shown that the replacement of clay and an iron-containing component with tefritobasalt and lead slag as part of the initial charge in the cement industry will lead to a decrease in temperature by 100 °C in the technological scheme of production and a reduction in energy consumption, since the theoretical specific consumption of raw materials is 1.481 t/t of clinker, which is approximately 70 kg lower than in traditional mixtures. The content of non-traditional components in total was 24.69%. In addition, tefritobasalts improved clinker formation processes, contributed to a decrease in the firing temperature, and intensified the clinker firing process. A small amount of lead slag (5.06%) introduced into the mixture changed the structure of the clinker and improved the process of mineral formation while also improving roasting and reducing the anthropogenic impact on the environment through the disposal of man-made waste. The strength of the experimental composite cements was tested after 7 and 28 days on small samples measuring $2 \times 2 \times 2$ cm. The physicomechanical characteristics and structure of composite cements were studied.

**Keywords:** industrial and technogenic waste; utilization; non-ferrous metals; environmental pollution; engineering; lead slag; clinker; composite materials; genesis; Portland cement

## 1. Introduction

The main raw materials in the production of cement are carbonate and clay rocks, as well as some types of industrial and technogenic waste [1–9]. Due to the depletion of stocks

of high-quality traditional raw materials, in the future, the maximum use of industrial man-made waste and the search for new types of raw materials that can be used in the production of Portland cement are required [10–19].

Igneous rocks such as basalts, tefritobasalts, etc. are non-traditional materials and industrial wastes that allow for a reduction in energy costs during the firing of cement clinker. It is also possible to completely replace the traditional clay component with coal mining waste from coal mines.

Studies on the use of basalts, tefritobasalts, lead slags, coal mining waste, etc. in the production of Portland cement as raw materials have been carried out [2,20–27]. In the technological cycle for the production of Portland cement clinker, the possibility of replacing clay with basalt as the main source of aluminosilicate substance has been shown. According to the results of physical and mechanical tests, the strength of cement obtained using basalt raw materials is not lower than conventional cement. The resulting clinker consists mainly of alite, belite, tricalcium aluminate, and ferrite. The effectiveness of using basalts to reduce the melting temperature of the mixture by 60–100 °C and accelerate the process of clinker formation has also been shown [28–34].

The possibility of using basalts as an additional cementing material in the composition of cement for wells has been shown [35–41].

The use of technogenic raw materials leads to significant changes in the content of heavy metals in clinker. This applies mainly to zinc, lead, and nickel. Studies have focused on the effect of zinc on the sintering process. It has been established that even the first percent of Zn content causes a decrease in the content of tricalcium aluminate in the clinker. Alloyed cements were at least as reactive as the reference cement [42–45].

The purpose of our research was to study the possibility of using tefritobasalt and lead slag as substitutes for clay and iron-containing components during clinker firing in the production of Portland cement. If implemented, this would be able to improve the state of the natural environment due to the utilization of man-made waste and conserve resources due to the involvement of waste instead of natural raw materials that need to be explored, mined, and maintained to maintain quarries [5,17,19,21,28]. Thus, there would also be an economic advantage to the proposed study [13,23].

## 2. Materials and Research Methods

To achieve the set goals, we studied the chemical and mineral compositions of raw materials and the clinker obtained from them. Scanning (raster) electron microscopic analysis of raw materials and baked clinkers was performed in the laboratory of local methods for studying substances at the Department of Petrology and Volcanology, Faculty of Geology, M.V. Lomonosov Moscow State University using an energy dispersive spectrometer (EDS) (INCA-Energy 350) based on a JEOL JSM-6480LV (Tokyo, Japan) microscope.

The content of the micro components in the initial material and products of the experiment was measured using the ICP-MS (Horiba, France) method at the Department of Geochemistry, M.V. Lomonosov Moscow State University. Samples for analysis were prepared by sintering with sodium carbonate. A 0.1 g sample was mixed with 0.3 g anhydrous sodium carbonate (Merck, Rahway, NJ, USA, Suprapur®) in an agate mortar and transferred to a corundum crucible. Sintering was carried out at 800 °C for one hour in a muffle furnace. The resulting sintered pellets were treated with 5 mL of $HNO_3$:HCl:HF (10:2:1), and after dissolution, diluted to 50 mL with deionized water (EasyPure®). For measurement, the solution was diluted with 3% nitric acid. The sintering method was tested and compared with multi-acid microwave decomposition of several standard samples [21].

The measurements were carried out on a high-resolution mass spectrometer with ionization in an inductively coupled plasma Thermo Element 2. Separation of ions was carried out using an analyzer with double focusing—magnetic and electrostatic modes. The ions were detected using an electron multiplier, which remained linear in the range from 1 to $1 \times 10^{10}$ ions per second. Calibration of the sensitivity of the instrument over the entire mass scale was carried out using reference 68-element solutions (ICP-MS-68A, HPS,

solutions A and B), which included all analyzed elements in the samples. To control the quality of measurements and take into account the drift in the sensitivity of the instrument, analyses of the samples were alternated with analyses of the reference sample with a frequency of 1:10. Indium at a concentration of 10 ppb was introduced into the samples as an internal standard. The analysis error ranged from 0.5 to 2 rel.%.

The X-ray diffraction data of the samples were obtained with a STOE-STADIMP (Darmstadt, Germany) powder diffract meter, with a curved Ge (111) monochromatic providing strictly monochromatic Co $K_{\alpha1}$-radiation. Data collection was carried out in the mode of stepwise overlapping of scanning areas from 6° to 90° by 2θ using a position-sensitive linear detector, the capture angle of which was 5° by 2θ, with a channel width of 0.02°. The phase composition was determined using the software package WinXPowSoftware // STOE&CIEGmbH, 2000 and Match! Software // CrystalImpactGbR, 2016, with their associated PDF-2 Powder Database (ICDD-2013).

### 2.1. Limestone of the Sastobe Deposit

The deposit is located in the Tyulkubas district of the Turkestan region (Kazakhstan). Limestone reserves are approximately 70 million tons. The productive stratum is composed of limestones of the Tournaisian Stage of the Carboniferous Age.

The bedlike limestone deposit has a northwest strike and a steep dip to the southwest. Its length is 1200 m, with a width of 320 m and a thickness of 670 m. The physical and mechanical properties of the limestones are bulk density—2.68–2.75 g/cm$^3$, water absorption, approximately—0.1–0.54%, loosening coefficient—1.24, and dry compression density—475–940 kg/cm$^2$. Limestones of the Sastobe deposit have special molecular strength and frost resistance [22]. The X-ray of limestone is shown in Figure 1.

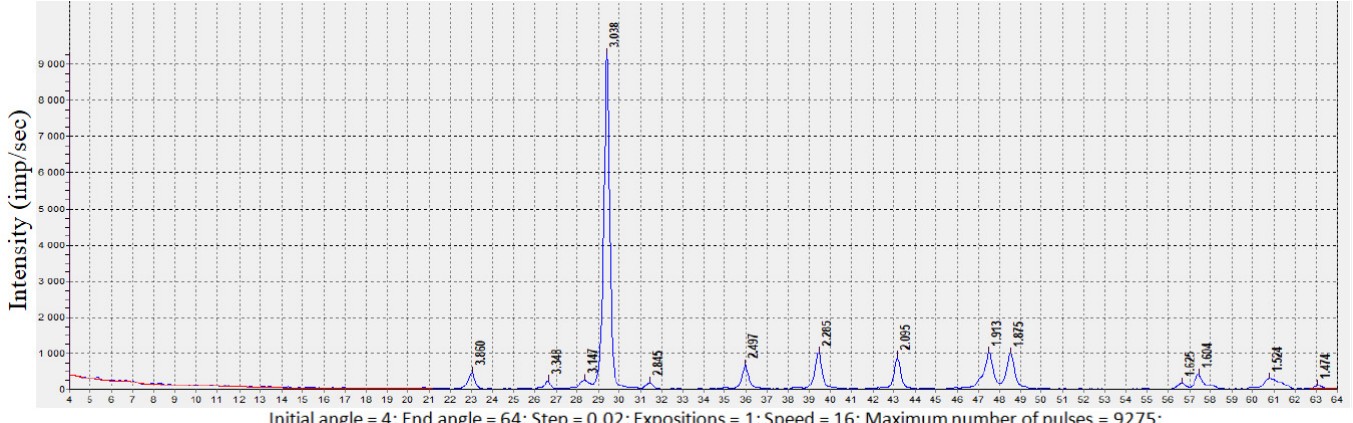

**Figure 1.** X-ray of limestone.

According to X-ray phase analysis, limestone consists mainly of calcite and quartz; the diffraction maxima of calcite were noted on the diffraction pattern: d = 3.86, 3.03, 2.49, 2.28, 2.09, 1.91, 1.87, and 1.60 Å. Quartz is also present: d = 3.34, 3.14, 2.84, 1.62, 1.52, and 1.47 Å. The content of oxides ($SiO_2$ and $Al_2O_3$) is low—2.67 and 0.26 wt.%, respectively, and the content of $Fe_2O_3$ is very low—0.57 wt.%. The content of MgO is insignificant—0.88 wt.%. The limestone is pure and highly basic, and the content of CaO is more than 52 wt.%. The content of alkalis is within the normal range (less than 0.54 wt.%).

### 2.2. Coal Mining Waste

Coal mines are located in the city of Lenger, Turkestan region (Kazakhstan). More than 6 million tons of waste have been accumulated in the storage facilities. The repository occupies approximately 25 hectares of land [23,24]. The X-ray of coal mining waste is shown in Figure 2.

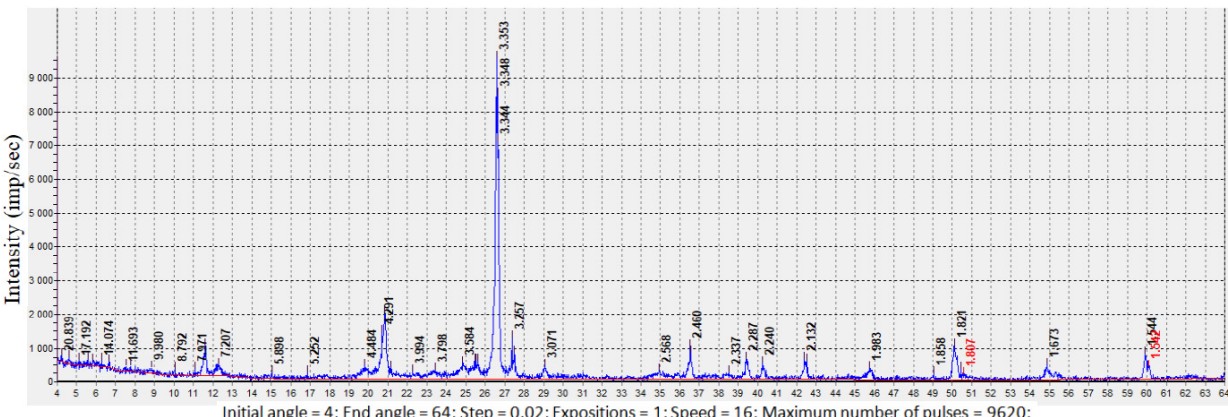

**Figure 2.** X-ray of coal mining waste.

According to the results of X-ray phase analysis, the coal mining wastes consist of kaolinite, gypsum, quartz, and a small amount of illite. Diffraction maxima were recorded: quartz d = 4.29, 3.35, 2.46, 2.28, 1.82, and 1.54 Å. There are also minerals of gypsum dihydrate d = 3.07, 2.24, 2.13, and 1.85 Å; calcite d = 2.28, 1.86, and 1.60 Å; dolomite d = 1.80, 1.54, and 1.11 Å; kaolinite clay minerals d = 7.207, 3.584, and 1.673 Å; illite (hydromica) d = 3.798, 3.257, and 1.983 Å; and carbon d = 2.132, 2.568, and 4.484 Å. The content of coal in the composition of coal mining waste was 15.23%. The content of silicon oxide was more than 55 wt.%, $Al_2O_3$—10.6 wt.%, and the carbon content in the coal mining waste was more than 15 wt.%. They can replace the aluminosilicate component in the raw mix.

*2.3. Tefritobasalt*

The Daubaba deposit is located in the Tyulkubas district of the Turkestan region (Kazakhstan). The deposit of Daubaba tefritobasalts has an inclined northeast direction. The length of the deposit is 2200 m, the width is almost 1200 m, and the thickness is from 13 to 70 m. The reserves of tefritobasalts are approximately 20 million tons. Tefritobasalts have a porphyritic structure (Figure 3).

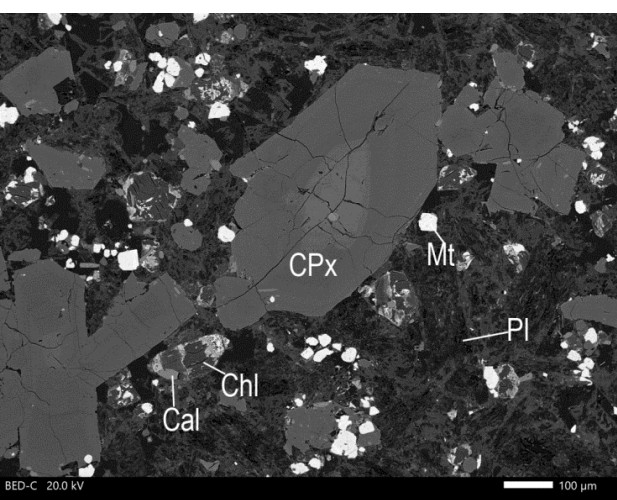

**Figure 3.** Micrograph of tefritobasalts. Legend: CPx—clinopyroxene, Chl—chlorite, Cal—calcite, Mt—magnetite, Pl—plagioclase relic.

Porphyritic phenocrysts are euhedral short-columnar grains of clinopyroxene. The composition of clinopyroxene corresponds to the isomorphic series diopside–hedenbergite. The grain size reaches 0.5 mm in length. It makes up approximately 30% of the total volume of the breed. Phenocrysts also include relics of plagioclase grains replaced by an aggregate

of secondary minerals. The predominant secondary mineral is zeolite (analcime). The relics have elongated prismatic outlines. Their size was 0.5–1 mm in length and approximately 0.1 mm in width. The content in the breed was approximately 15%. Plagioclase was almost completely replaced. The exact composition of the original mineral cannot be determined.

The sample contained isometric grains of magnetite ranging in size from hundredths of a millimeter to ≈0.2 mm. It contained impurities of Al, Mn, V, Si, Cr, and Zn. The amount of magnetite in the rock did not exceed 15%. Small (up to 0.1 mm in size) grains of chlorite, calcite, alkali feldspar, and apatite were also found. Their total content did not exceed 15%. The X-ray of tefritobasalt is shown in Figure 4.

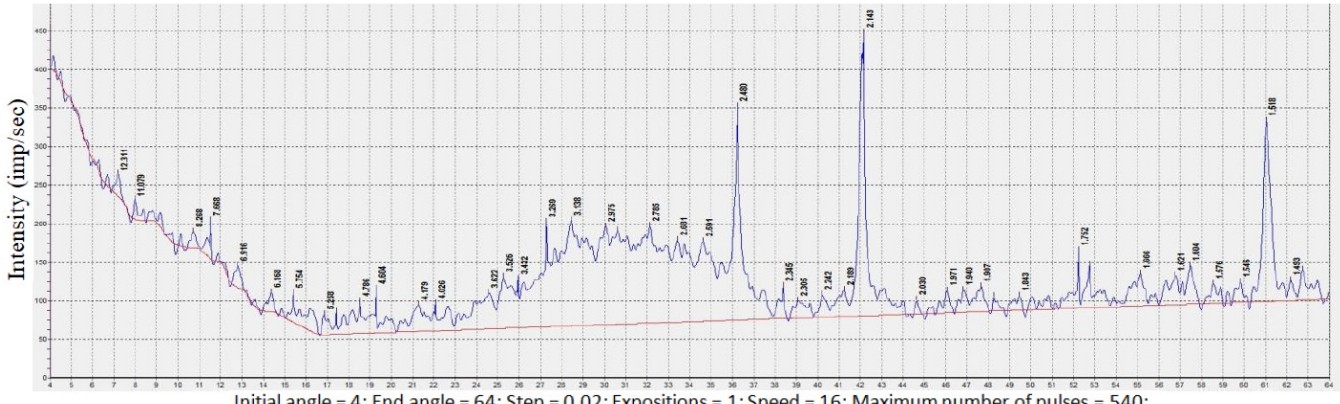

**Figure 4.** X-ray diffraction pattern of tefritobasalt from the Daubaba deposit.

The diffraction peaks of the following minerals were noted on the X-ray pattern of tefritobasalt: pyroxene d = 3.62, 3.26, 2.78, 2.30, 2.19, 1.97, and 1.94 Å; plagioclase d = 4.02, 3.43, 2.68, 2.59, and 2.14 Å; biotite d = 4.78, 4.60, 3.52, 1.66, 1.60, and 1.54 Å; anorthite d = 4.17, 3.13, 2.24, 1.90, 1.84, and 1.51 Å; and olivine d = 2.97, 2.48, 2.34, 2.03, 1.75, 1.62, 1.57, and 1.49 Å. According to the results of the X-ray phase analysis, olivine and clinochlore were also found in the rock in small amounts up to 5%. The groundmass was composed of a submicron aggregate, the mineral composition of which, apparently, was close to that of phenocrysts.

The temperature at the beginning of the melting of tefritobasalts, determined by the dilatometric method, was 1280 °C. At 1450 °C, the rock powder (sifted through a 02 sieve) completely melted, clarified, and homogenized within 45 min [25]. The physical and mechanical properties of the tefritobasalts were as follows: density = 2.0 g/cm$^3$ and compressive strength = 147.6–195.8 MPa. According to chemical analysis, tefritic basalt contains $SiO_2$—45.54 wt.%, $Al_2O_3$—more than 10 wt.%, and $Fe_2O_3$—approximately 8.5 wt.%. The magnesium content was 6.95 wt.%. It contained a significant amount of alkalis ($K_2O + Na_2O$), which was 6.54 wt.%.

### 2.4. Lead Slag

The slag plant used was JSC "Yuzhpolymetal", which is located in the city of Shymkent (Kazakhstan). The balance reserves of lead slags are approximately 2 million tons [26,27]. Lead slags contain up to 37.25% $Fe_2O_3$ and can replace the corrective additive. In addition, lead slags contain up to 14% CaO and partially replace the carbonate component. The X-ray of lead slag is shown in Figure 5.

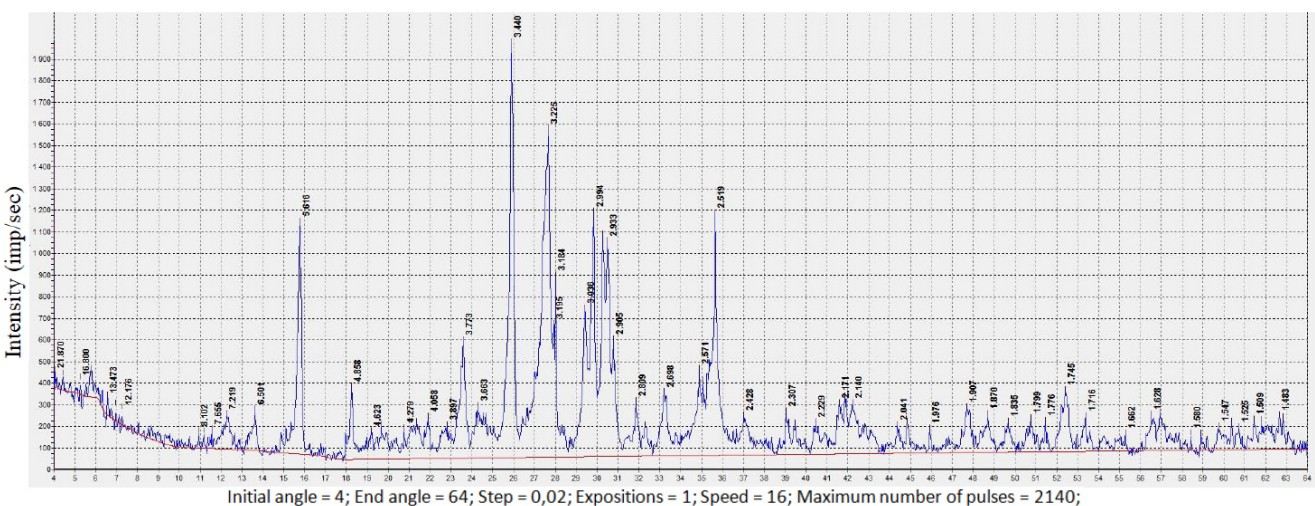

**Figure 5.** X-ray of lead slag.

According to the results of the X-ray phase analysis of lead slag, the diffraction maxima were recorded: fayalite d = 3.77, 2.81, 2.31, 1.77, and 1.58 Å; wustite d = 4.27, 2.14, 1.52, and 1.50 Å; hematite d = 3.66, 2.69, 2.42, 2.22, 1.83, 1.66, and 1.48 Å; melilite d = 2.51, 2.04, 1.87, 1.74, 1.71, and 1.54 Å; and zinc spinel d = 1.90 and 1.62 Å.

According to the microprobe study, lead production slag has an incompletely crystalline porphyry structure (Figure 6). The bulk was composed mainly of glass. The amount of glass was approximately 60% of the total volume of the sample. The structure of the ground mass was heterogeneous. It alternated areas with vitrophyric and hyalopilitic structures. In the most crystallized areas, an increase in the content of zinc and sulfur and a decrease in the amount of calcium were noted.

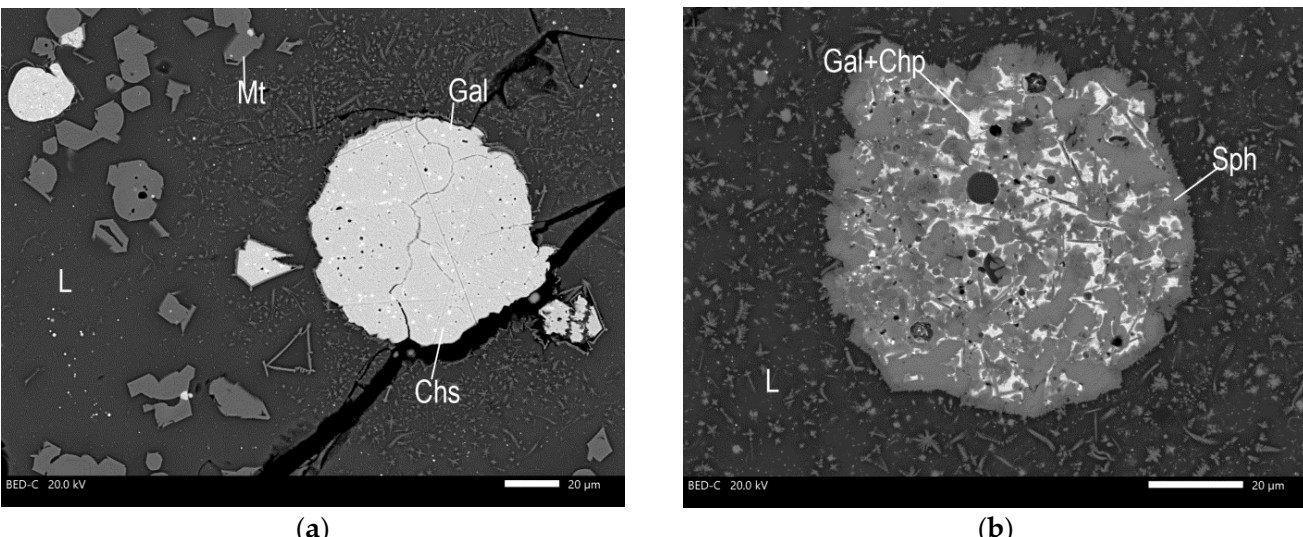

**Figure 6.** Micrographs of lead slag with various types of phenocrysts. Symbols: (**a**) Gal—galena, Chs—chalcocite, Mt—magnetite, (**b**) Chp—chalcopyrite, and Sph—sphalerite.

Phenocrysts are represented by various sulfides that form rounded isometric aggregates. Their size varied from a few microns to ≈0.5 mm. The mineral composition was different. There were aggregates composed of approximately 95% chalcocite ($Cu_2S$) and 5% galena (PbS), and aggregates composed mainly of sphalerite (ZnS) with a small amount (up to 5%) of chalcopyrite ($CuFeS_2$) and galena. Subhedral grains of hematite ($Fe_2O_3$) were

present as phenocrysts. Their average size was 10 μm. The total amount in the sample was approximately 20%.

Based on the data on the chemical composition of the raw materials in the ROCS program [28], a four-component calculation of the composition of the raw batch was made, consisting of "Limestone, tefritobasalt, coal mining waste, and lead slag". For the optimal modular characteristics of the resulting clinkers, the quantitative ratio of the initial components, as well as the calculated chemical and mineral composition of the clinker, were determined.

The raw materials were crushed and sieved through a No. 008 sieve. As a result, the surface area of particles per 1 g of material (specific surface area) was 3118–3326 cm$^2$/g, and the average particle size was 5.67–5.92 μm. In accordance with the obtained calculated composition, a mixture was made from them. This mixture, on a hydraulic press under a pressure of 20 MPa, was molded into tablets with a diameter of 15 mm and a height of 10 mm.

The firing of the tablets was carried out in an electric laboratory furnace, SX-2–18TP (NanYang, China), at M. Auezov South Kazakhstan University (Kazakhstan). The temperature regime of firing fully corresponded to the temperature regime of a rotary industrial kiln. The rise in temperature to 1350 °C occurred within 2–2.5 h. Exposure at the maximum temperature was carried out for 30 min. A general view of the furnace, unfired pellets, and clinkers obtained at a temperature of 1350 °C is shown in Figure 7.

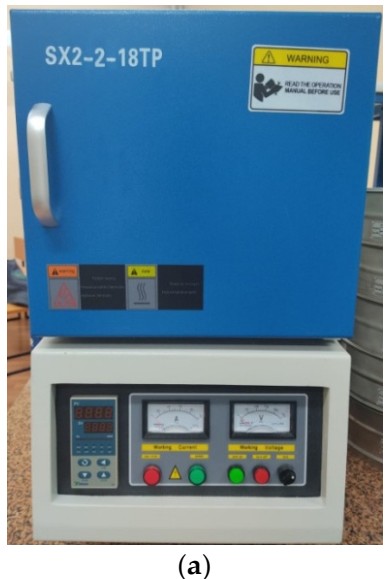
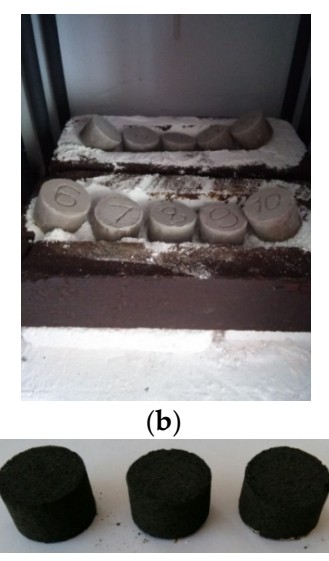

**(a)** **(b)** **(c)**

**Figure 7.** General view of the furnace (**a**), unfired pellets (**b**), and clinkers (**c**), obtained at a temperature of 1350 °C.

The duration of clinker firing was controlled by the degree of absorption of CaO into the clinker minerals. The burned tablets were crushed. The qualitative content of free CaO was determined microscopically. The quantitative content of CaO in the clinkers was determined using the ethylene–glycerate method [29].

## 3. Results and Discussion

In accordance with the results of calculating the composition of the raw mixture (Table 1), the optimal characteristics of the clinker for these initial conditions were as follows: SC = 0.94; silicate module $n$ = 2.02; and alumina modulus $p$ = 0.95. The saturation factor (SC) of the clinker is the ratio of CaO remaining after saturation of $Al_2O_3$ and $Fe_2O_3$ to $3CaO \cdot Al_2O_3$ and $4CaO \cdot Al_2O_3 \cdot Fe_2O_3$ to the amount of CaO that is necessary to completely saturate the silica to $3CaO \cdot SiO_2$. The SC value of the factory clinkers varied from 0.85 to 0.95, depending on the composition and properties of the raw materials, the firing conditions, etc. The silicate module is the ratio of the $SiO_2$ content to the sum of the

oxides $Al_2O_3$ and $Fe_2O_3$. The $n$ module value ranged from 1.7 to 4.0. The alumina modulus is the ratio of the $Al_2O_3$ content to the $Fe_2O_3$ content and characterizes the composition of the aluminoferrite phase in the clinker. The values of the $p$ modulus ranged from 0.9 to 3.0. The theoretical specific consumption of raw materials is 1.481 t/t of clinker, which is approximately 70 kg lower than in traditional mixtures. The content of non-traditional components in the mixture was 24.69% (Table 2).

**Table 1.** Chemical composition of raw materials, waste, and igneous rocks.

| Name | The Chemical Composition, mas. % | | | | | | | | | | | | | | | | |
| | $SiO_2$ | $Al_2O_3$ | $Fe_2O_3$ | CaO | MgO | $SO_3$ | $Na_2O$ | $K_2O$ | $TiO_2$ | Cl | $Cr_2O_3$ | ZnO | PbO | CuO | Loss of Ignition | Other | Total |
|---|---|---|---|---|---|---|---|---|---|---|---|---|---|---|---|---|---|
| Limestone | 3.87 | 1.04 | 0.57 | 52.83 | 0.88 | 0.10 | - | 0.12 | 0.018 | 0.02 | 0.012 | - | - | - | 40.20 | 0.37 | 100.0 |
| Tefrito basalt | 45.54 | 10.70 | 8.53 | 10.66 | 6.95 | 0.20 | 4.04 | 2.50 | 0.91 | 0.017 | 0.007 | - | - | - | 5.37 | 4.57 | 100.0 |
| Coal mining waste | 55.50 | 10.60 | 2.01 | 3.21 | 0.70 | 0.79 | - | 2.35 | 0.38 | - | - | - | - | - | 24.08 | 0.38 | 100.0 |
| Lead slag | 25.94 | 6.44 | 37.25 | 14.71 | 6.15 | 0.04 | 1.24 | 1.36 | 0.35 | 0.006 | 0.063 | 4.34 | 0.52 | 0.94 | 0.65 | - | 100.0 |

**Table 2.** Results of calculating the composition of the raw batch and the resulting clinker.

| Materials | $SiO_2$ | $Al_2O_3$ | $Fe_2O_3$ | CaO | MgO | $SO_3$ | L.W.C | Other |
|---|---|---|---|---|---|---|---|---|
| Limestone Sastobe | 3.87 | 1.04 | 0.57 | 52.83 | 0.88 | 0.10 | 40.71 | - |
| Tefritobasalt Daubaba | 45.54 | 10.70 | 8.53 | 10.66 | 6.95 | 0.20 | 7.92 | 9.50 |
| Coal mining waste from the Lenger mines | 55.50 | 10.60 | 2.01 | 3.21 | 0.70 | 0.79 | 24.08 | 3.11 |
| Lead slag of Yuzhpolimetall JSC plant | 25.94 | 6.44 | 37.25 | 14.71 | 6.15 | 0.04 | 0.10 | 9.37 |

| By component chemical composition of the raw batch | | | | | | | | | Component content | |
| | | | | | | | | | kg/kg clinker | % |
|---|---|---|---|---|---|---|---|---|---|---|
| Limestone | 2.91 | 0.78 | 0.43 | 39.78 | 0.66 | 0.075 | 30.65 | - | 1.1160 | 75.31% |
| Tefritobasalt | 9.20 | 2.16 | 1.72 | 2.15 | 1.41 | 0.040 | 1.61 | 1.92 | 0.1575 | 10.63% |
| Coal mining waste | 0.58 | 0.11 | 0.02 | 0.04 | 0.01 | 0.008 | 0.25 | 0.033 | 0.1575 | 10.63% |
| Lead slag | 0.88 | 0.22 | 1.27 | 0.51 | 0.21 | 0.001 | 0.003 | 0.321 | 0.0508 | 3.43% |
| Raw mix | 13.59 | 3.28 | 3.45 | 42.48 | 2.29 | 0.13 | 32.52 | 2.27 | 1.4818 | 100.00% |

| By component chemical composition of the clinker | | | | | | | | | Component content | |
|---|---|---|---|---|---|---|---|---|---|---|
| Limestone | 4.32 | 1.16 | 0.64 | 58.95 | 0.98 | 0.112 | - | - | 66.17% | |
| Tefritobasalt | 13.63 | 3.20 | 2.55 | 3.19 | 2.08 | 0.060 | - | 2.845 | 14.38% | |
| Coal mining waste | 0.86 | 0.17 | 0.03 | 0.05 | 0.01 | 0.012 | - | 0.049 | 14.39% | |
| Lead slag | 1.31 | 0.33 | 1.89 | 0.75 | 0.31 | 0.002 | - | 0.476 | 5.06% | |
| clinker | 20.14 | 4.86 | 5.11 | 62.95 | 3.39 | 0.19 | - | 3.37 | 100.00% | |

| Chemical composition of raw mix and clinker | | | | | | | | | SC | $n$ | $p$ | TEC. (kcal/kg) | $G_{fuel}$. kg ref.fuel/t cl) |
|---|---|---|---|---|---|---|---|---|---|---|---|---|---|
| Raw mix | 13.59 | 3.28 | 3.45 | 42.48 | 2.29 | 0.13 | 32.52 | 2.27 | 0.94 | 2.02 | 2.02 | - | - |
| clinker | 20.14 | 4.86 | 5.11 | 62.95 | 3.39 | 0.19 | - | 3.37 | 0.94 | 0.95 | 0.95 | 364.1 | 196 |

| Mineralogical composition of the clinker | | | | | | |
|---|---|---|---|---|---|---|
| Minerals | $3CaO \cdot SiO_2$ | $2CaO \cdot SiO_2$ | $3CaO \cdot Al_2O_3$ | $4CaO \cdot Al_2O_3 \cdot Fe_2O_3$ | $CaSO_4$ | MgO |
| wt.% | 57.88 | 18.82 | 6.46 | 11.61 | 0.32 | 3.39 |

Symbols: $C_3S$—alite ($3CaO \cdot SiO_2$), $C_2S$—belite ($2CaO \cdot SiO_2$), $C_3A$—tricalcium aluminate ($3CaO \cdot Al_2O_3$), $C_4AF$—tetracalcium aluminoferrite ($4CaO \cdot Al_2O_3 \cdot Fe_2O_3$), LOI—loss on ignition.

Figure 8 shows the dependence of the content of alite ($3CaO \cdot SiO_2$) in the clinker on the value of SC and the silicate and alumina modules of the raw batch. With an increase in the SC from 0.8 to 0.95 and an $n$ modulus from 1.5 to 3.5, the alite content increased from 40% to 70%. With the silicate modulus $n = 2.02$, the value of the alumina modulus was $p = 0.95$. An increase in the silicate modulus contributed to an increase in the alumina modulus as well as an increase in the content of tricalcium aluminate. According to the results

of calculating the composition of the raw charge consisting of "Limestone, tefritobasalt, coal mining waste, and lead slag" is suitable for producing sulfate-resistant Portland slag cement clinkers of the CEMIII/AC grade, according to GOST 22266-2013 [30]. The calculated chemical and mineral composition of the clinker contained no more than 7% of $3CaO \cdot Al_2O_3$, and the amount of $Al_2O_3$ and MgO did not exceed 5%. The total content of $3CaO \cdot Al_2O_3 + 4CaO \cdot Al_2O_3 \cdot Fe_2O_3$ was 18.07% and did not exceed the maximum allowable values ($3CaO \cdot Al_2O_3 + 4CaO \cdot Al_2O_3 \cdot Fe_2O_3 < 22\%$). Alite in the composition of the clinker was 57.88%, and in belite ($2CaO \cdot SiO_2$) it was 18.82%.

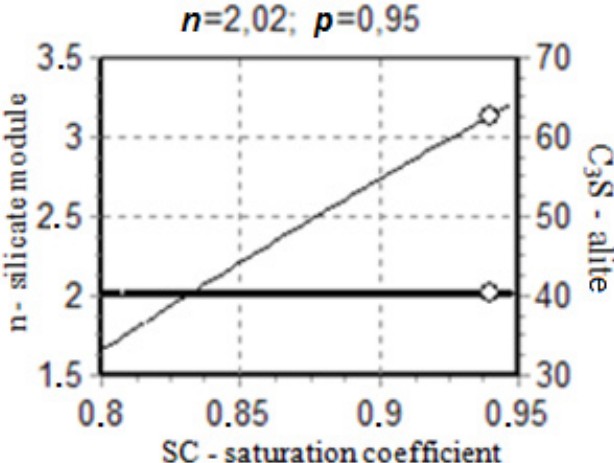

**Figure 8.** Dependence of the content of alite ($C_3S$) in the clinkers on the value of modules (*n* and *p*) and the saturation coefficient in the raw mixtures.

Clinker firing is characterized by a complex physical and chemical process. The reactions occurring in the process of clinker formation determine the quality of the finished product and its phase composition.

Clinker fired at a temperature of 1350 °C reached almost complete assimilation of calcium oxide within 30 min, and the content of free CaO in the resulting clinker was 0.2% (Table 3). The quality of the clinker was high, and the chemical and mineralogical composition met the requirements. It has been established that in the studied raw mixtures, the process of clinker formation ends at 1350 °C (i.e., 100 °C lower than in traditional raw mixes) [31–40].

**Table 3.** Specific consumption of raw materials and the amount of free CaO in the clinker.

| Mix | Composition of the Raw Batch, wt.% | | | | Specific Consumption of the Raw Materials, t/t of Clinker | | | | SC | Modules | | Amount of Free CaO at 1350 °C, % |
|---|---|---|---|---|---|---|---|---|---|---|---|---|
| | Lime Stone | Tefrito Basalt | Coal Mining Waste | Lead Slag | Lime Stone | Tefrito Basalt | Coal Mining Waste | Lead Slag | | *n* | *p* | |
| 1 | 66.17 | 14.38 | 14.39 | 5.06 | 1.1160 | 0.1575 | 0.1575 | 0.0508 | 0.94 | 2.02 | 0.95 | 0.2 |

On the radiograph of the clinker obtained by burning at a temperature of 1350 °C of energy and resource-saving raw mixtures, lines characteristic of free CaO are found (Figure 9).

When burning the energy resource-saving raw material mixture at 1350 °C, the following clinker minerals were formed: alite ($C_3S$) d = 2.19, 2.62, 2.78, and 3.05 Å; belite ($C_2S$) d = 1.73, 1.93, 2.74, and 2.88 Å; tricalcium aluminate ($C_3A$) d = 1.87, 1.90, 2.95, and 4.04 Å; four-calcium aluminoferrite ($C_4AF$) d = 2.03, 2.13, and 2.66 Å The peaks characteristic of free CaO are absent.

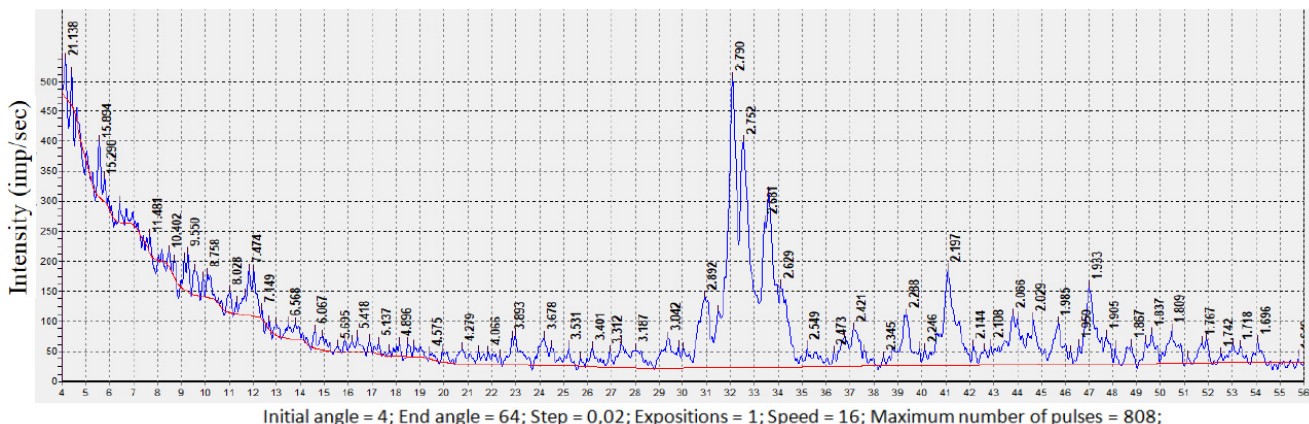

**Figure 9.** Radiographs of the clinkers.

Figure 10 shows micrographs of the polished section of the resulting clinker. The synthesized clinker had a full-crystalline structure and a massive texture. The main crystalline phases were evenly distributed over the sample [32–43]. Alite was represented by isometric subhedral crystals 20–50 μm in size. Belite inclusions were observed in large rhombohedral alite crystals. The content of alite in the composition of the clinker was 57.88%. Belite was represented by small grains up to 10 μm in size. The crystals had a round, near-isometric shape. Belite grains were in direct contact with alite grains, which indicates their formation by reactions in the solid state. The belite content in the clinker composition was 18.82%.

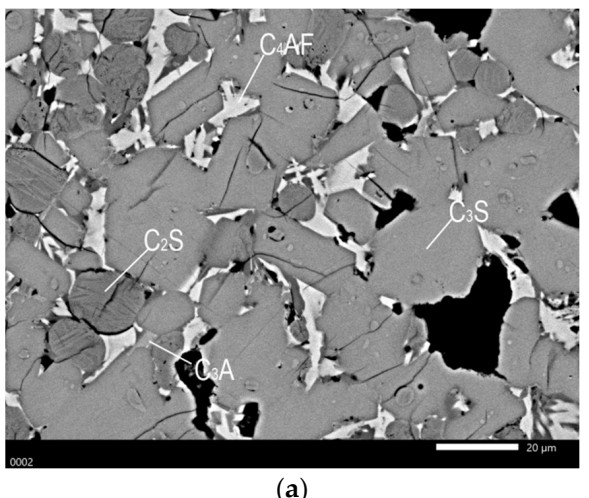

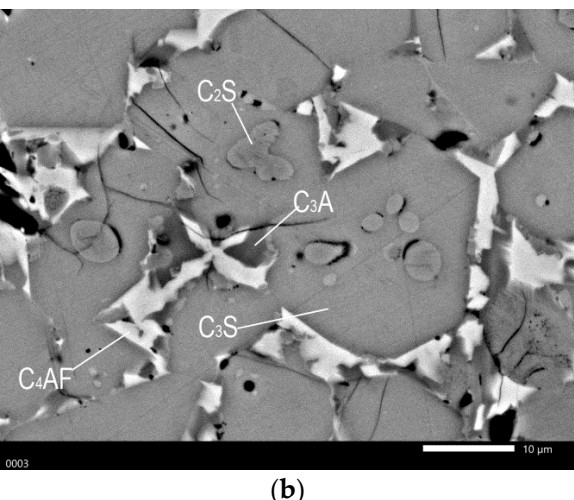

(**a**)                                                (**b**)

**Figure 10.** Micrographs of the clinker (**a**)—there was an aluminate phase a dark intermediate substance, (**b**)—and an aluminoferrite phase—a light intermediate substance.

Along the grain boundaries of alite and belite, there was an aluminate phase—a dark intermediate substance—and an aluminoferrite phase—a light intermediate substance. The intermediate substance was present in sufficient quantity ($3CaO \cdot Al_2O_3$ + $4CaO \cdot Al_2O_3 \cdot Fe_2O_3$ = 18.07%). In traditional clinker, the aluminoferrite phase is 14%. The aluminoferrite phase in all parameters (hydraulic activity, hardening rate, and strength characteristics) occupied an intermediate position between $3CaO \cdot SiO_2$ and $2CaO \cdot SiO_2$. Hardening products have increased water and corrosion resistance. In clinker sinters based on tefritobasalt, well-formed, regular-shaped alite crystals are formed. Tefritobasalt improved and accelerated the processes of clinker formation, helped reduce the firing temperature (the beginning of melting was 1280 °C), and intensified the process of firing the clinker. The traditional raw mix was heated to clinker sintering at 1450–1500 °C.

A small amount of lead slag (5.06%) introduced into the mixture changed the structure of the clinker and improved the process of mineral formation. The addition of lead slag to the raw mix improved fire ability and calcium silicate formation at a lower temperature. Zinc oxide contained in the composition of the slag contributed as a mineralizer to the dissolution of the residue of free lime at 1350 °C. Its presence affected the course of solid reactions and the formation of silicates and calcium aluminates. During the firing process, the material lost a significant amount of these metals. In the finished product, their content was low. According to the ICPMS, the content of lead in the clinker was 41 ppm, and zinc was 197 ppm.

The microstructure of traditional cement clinker is uniformly granular and finely crystalline. The optimal size of alite crystals is 20–40 microns. The content of tricalcium aluminate was approximately 8%. The saturation coefficient of the raw mixture was in the range of 0.90–0.92.

The conducted research shows the possibility of replacing traditional raw materials with man-made industrial waste such that economically beneficial phenomena, including resource conservation and utilization of man-made waste, are possible, while reducing the anthropogenic impact on the region.

In a laboratory ball mill, cement was mixed with 5% gypsum. The residue on sieves No. 02 and No. 008 was determined every 10 min, and the total grinding time was 30 min. A study of the kinetics of grinding showed that the burning temperature and the amount of slag introduced into the raw material mixture had a noticeable effect on the grinding process of cements. After 30 min of grinding, the cement residue on sieve No. 008 was 13%. The specific surface of the cements was determined using a PSX-K. According to the results of the analysis, the specific surface of the cements was 3245 cm$^2$/g, and the average particle size was 5.97 microns.

In the laboratory, the physicomechanical properties of the experimental Portland cement were tested in small samples of 2 × 2 × 2 cm after 7 and 28 days [41–46]. As can be seen from the data in Table 4, the tensile strength of the obtained cements increased with increasing hardening time. After 28 days, the compressive strength of the cements was 41.8 MPa. These indicators correspond to the cement grade for strength M400 according to GOST 10178-85 or strength class 42.5 according to GOST 31108-2016.

**Table 4.** Physicomechanical properties of experimental Portland cement.

| Cement | Sieve Residue,% | | Specific Surface Area, cm$^2$/g | Strength of Small Samples 2 × 2 × 2 cm, (MPa) | |
| --- | --- | --- | --- | --- | --- |
| | No. 02 | No. 008 | | 7 Day | 28 Day |
| Cement | 3.3 | 13 | 3245 | 27.4 | 41.8 |

These studies confirm and supplement previously conducted similar studies by both international scientific centers and scientific communities [31–33,36,38,43–46].

## 4. Conclusions

1.  The conducted research shows the possibility of replacing traditional raw materials with man-made industrial waste such that economically beneficial phenomena, including resource conservation and utilization of man-made waste, are possible, while reducing the anthropogenic impact on the region.
2.  In the studied initial experimental compositions, the replacement of the clay- and iron-containing component with tefritobasalt and lead slag led to a decrease in the melting temperature of the mixture by 100 °C relative to traditional compositions and is environmentally and economically feasible.
3.  In the samples obtained from the four-component raw material mixture "Limestone—tefritobasalt—coal mining waste—lead slag", it is clear that the synthesized clinker has a full-crystalline structure with a content of 57.88% alite and 18.82% belite.

4.  According to the calculation results, SC = 0.94, silicate modulus *n* = 2.02, and alumina modulus *p* = 0.95. The theoretical specific consumption of raw materials is 1481 t/t of clinker, which is approximately 70 kg lower than in traditional mixtures. The content of non-traditional components in total was 24.69%.

5.  According to the results of chemical analysis, the content of free CaO in the clinker was 0.2%, which meets the requirements of GOST. The quality of the synthesized clinker was high, the minerals were formed correctly, and the size of the alite crystals was 20–50 microns.

6.  The strength of the obtained Portland cement after 28 days under compression was 41.8 MPa, which corresponds to the cement grade of strength M400 according to GOST 10178-85 or strength class 42.5 according to GOST 31108-2016.

**Author Contributions:** Conceptualization, N.Z. and O.K.; methodology, B.S.; software, A.A.; validation, Y.A., A.B. and S.S.; formal analysis, L.N.; investigation, F.N.; resources, G.S.; data curation, Y.Z.; writing—original draft preparation, P.K.; writing—review and editing, A.I.; visualization, O.K.; supervision, N.Z.; project administration, P.K.; funding acquisition, N.Z. All authors have read and agreed to the published version of the manuscript.

**Funding:** The study is funded by the authors.

**Institutional Review Board Statement:** Not applicable.

**Informed Consent Statement:** Not applicable.

**Data Availability Statement:** Data sharing is not applicable to this article.

**Acknowledgments:** This research was supported by the JSC Center for International Programs "Bolashak" of the Ministry of Science and Higher Education of the Republic.

**Conflicts of Interest:** The authors declare no conflict of interest.

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
