# Peer review of "Studies of Utilization of Technogenic Raw Materials in the Synthesis of Cement Clinker from It and Further Production of Portland Cement"

_jcs, doi:10.3390/jcs7060226_

Round 1
Reviewer 1 Report
I made some comments directly on the paper!

Author Response
Hello, dear Reviewer!
Thank you for your high appreciation and thorough review of our article.
We answer the question asked for you.
Question 1. in oven ?
Answer 1. The experiment was carried out at atmospheric pressure and a temperature of 1350 °C in an electric furnace.
Question 2. he melting temperature for each individual component was somehow studied, so that an analysis can be made from this point of view as well. It is very likely that these different temperatures contributed to your conclusion that the melting temperature drops by about 100 oC.
Answer 2. The temperature of the beginning of melting of tephritobasalts, determined by the dilatometric method, was 1280 °C. Tefritobazalt, introduced in an amount of 10.63%, accelerates the processes of mineral formation due to the appearance of a liquid clinker phase at low temperatures. Lead slag is introduced in an amount of 3.43%. The mineralizing effect of zinc oxide contained in lead slag makes it possible to complete the clinker formation processes at 1350 oC. In general, this leads to an acceleration of firing processes and the completion of clinker formation at temperatures of 100 oC.
Question 3. Describe what Table 1 represents!
Answer 3. The chemical composition of raw materials, waste and igneous rocks. The chemical composition was determined according to GOST 5382-2019 Cements and cement production materials. Methods of chemical analysis.
Question 4. Make the text more readable!
Answer 4. Thank you, I agree with the comment and corrected.
These data are the range of the X-ray machine (intensity (iml/sec), initial angle, final angle, step, exposure, speed, maximum number of pulses). The inscriptions are very small to translate them from Russian into English and it is impossible to insert them as a drawing, so I removed them.
Question 4. Space !
Answer 4. The intensity of the line is denoted by - d.
Question 5. Why such a wide field? There is an explanation!
Answer 5. Thank you. The remark has been corrected. The number 1 remained unwritten. Compressive strength is 147.6-195.8 MPa.
Question 6. The figures and the text on them should be clearer. Try another method of retrieving the graphics from the device!
Answer 6. Thank you. The remark has been corrected.
Question 7. Make comments about Table 2!
Answer 7. Table 2 shows the results of calculating the composition of the raw charge and the resulting clinker. The calculation was carried out using the ROСS program. The main indicators of the clinker obtained are determined.
Question 8. What happens if 1450oC is still reached in the process of melting the raw materials? What would be the quality of the clinker in this situation?
What differences can appear compared to the clinker obtained at 1450oC?
Do you have a comparison in this sense?
Answer 8. If we have developed the composition of the raw charge, the clinker formation process ends at a temperature of 1350 oC, but we continue to burn up to 1450 oC, this will be clinker burnout. Clinker burnout is a long exposure at the highest temperature. After almost all the lime has been absorbed, it reduces the strength of cement by 35% compared to the strength of clinker cement fired according to the optimal regime. The decrease in strength during burnout is explained by the fact that in this case, after the practical cessation of the process of assimilation of lime and other reactions, the stabilization of the formed compounds occurs, the growth of crystals and the reduction of defects in crystal lattices. Clinker burnout is rare. Such clinker is very difficult to grind, and, in addition, excessive fuel is spent on firing it.
Reviewer 2 Report
Please find the comments in the attachment.

Author Response
Good afternoon, dear Reviewer!
Thank you so much for your valuable comments and suggestions that contribute to the improvement of our manuscript.
We have finalized the document slightly improving it. But, unfortunately, the study of hydration issues was not part of the task of our research. This will be the next stage of our research, and we will write another scientific article on them.
In the text, we indicated additional environmental benefits in the processing of man-made waste.
We have replaced the title of the article, according to your suggestion. Thank you.
Unfortunately, due to our material and technical equipment, we were unable to consider in more detail the issue of the combined mineralizing effect of all indirect secondary elements introduced during sintering of clinker and secondary raw materials (minerals, waste, by-products). -products, etc.). Since the elements acting as mineralizers are contained in minimal quantities, and the available physico-chemical equipment at our disposal cannot quite recognize these elements and compounds.
Added and made a comparison.
Unfortunately, we do not have such equipment. And we have to use what we have in our universities.
References have been replaced
1. Since the elements acting as mineralizers are contained in minimal quantities, and the available physico-chemical equipment cannot quite recognize these elements and compounds. In this regard, we do not have the opportunity to conduct a SEM-EDXS study for elemental mapping of all minor cations.
2. This will be the next stage of our research and we will write another scientific article on them.
3. Since the minor elements acting as mineralizers are contained in minimal quantities, and the available physico-chemical equipment at our disposal cannot quite recognize these elements and compounds.
4. Unfortunately, the study of hydration issues did not go into the task of our research. This will be the next stage of our research and we will write another scientific article on them.
The list of references has been replaced.
All changes are highlighted in color yet
Thank you again for the work done and for your valuable comments. Please be understanding. Since we do not have to work with the most advanced equipment in the field of physico-chemical research.
We wish you health, creative success, prosperity and new scientific achievements.
Sincerely, the team of Authors.
Round 2
Reviewer 1 Report
The work could have looked better, but it's okay!